# Accelerating Search-Based Planning for Multi-Robot Manipulation by Leveraging Online-Generated Experiences

**Primary Keywords:** *(3) Robotics; (7) Multi-Agent Planning;*

## Abstract

An exciting frontier in robotic manipulation is the use of multiple arms at once. However, planning concurrent motions is a challenging task using current methods. The high-dimensional composite state space renders many well-known motion planning algorithms intractable. Recently, multi-agent path finding (MAPF) algorithms have shown promise in discrete 2D domains, providing rigorous guarantees. However, widely used conflict-based methods in MAPF assume an efficient single-agent motion planner. This poses challenges in adapting them to manipulation cases where this assumption does not hold, due to the high dimensionality of configuration spaces and the computational bottlenecks associated with collision checking. To this end, we propose an approach for accelerating conflict-based search algorithms by leveraging their repetitive and incremental nature – making them tractable for use in complex scenarios involving multi-arm coordination in obstacle-laden environments. We show that our method preserves completeness and bounded sub-optimality guarantees, and demonstrate its practical efficacy through a set of experiments with up to 10 robotic arms.

## Introduction

The synchronous use of multiple robotic arms may enable new application domains in robotics and enhance the efficiency of tasks traditionally carried out by a single arm. For example, in pick-and-place tasks, multiple arms can potentially be more efficient than a single one, and in a manufacturing setting, multiple arms can be used to assemble a product collaboratively, unlocking the capability to perform tasks that are beyond the scope of a single arm. However, the inherent complexity of single-agent motion planning for robot manipulation (Canny 1988) makes it challenging to plan for multiple arms while ensuring collision-free paths, and thus has left the Multi-Robot-Arm Motion Planning (M-RAMP) problem a relatively under-explored frontier in robotics.

To enable the use of multiple arms in more complex scenarios, we propose a method for accelerating multi-robot-arm motion planning. Our approach capitalizes on a key observation: widely-used multi-agent path-finding algorithms exhibit a significant degree of repetitive planning. We exploit this repetitiveness by developing an approach that leverages experiences gathered during the planning process. Unlike previous approaches that utilize incremental search techniques (Boyarski et al. 2021), we allow the use of bounded

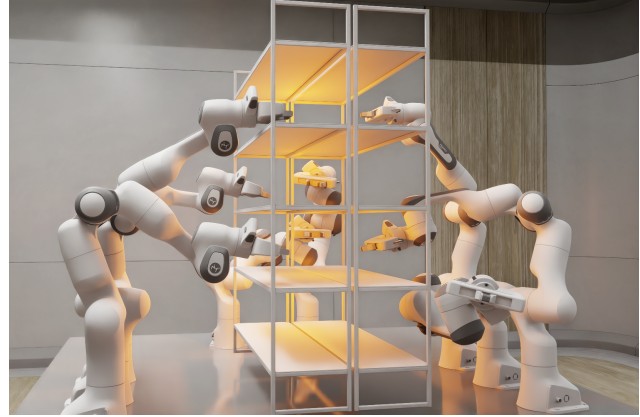

Figure 1: A team of 8 robotic manipulators, each of 7-DOF, collaborating in a shelf-rearrangement task. Planning concurrent motions for all arms requires a motion planner capable of efficiently exploring the high-dimensional state space of a single arm, and also reasoning about the motions of multiple robots operating in the shared task space.

sub-optimal search techniques, which are crucial for exploring high-dimensional state spaces. To this end, we accelerate the single-agent planning process by reusing online-generated path experiences to speed up multi-agent search, ensuring both completeness and solution quality guarantees

Our contributions in this paper are threefold. First, we introduce a novel method for multi-robot-arm motion planning. Second, we provide a comprehensive theoretical analysis of our proposed framework, demonstrating its bounded sub-optimality guarantees. Third, we offer an empirical evaluation of our method and other algorithms in various multi-arm manipulation scenarios that include deadlock avoidance, cluttered environments, and closely interacting goals.

## Related Work

The literature has extensively examined path planning for both single and multi-agents. In the context of single-agent search, decades of research have yielded algorithms capable of scaling successfully to high-dimensional and computationally expensive search spaces. However, efforts in multi-agent path planning have generally been applied to domains

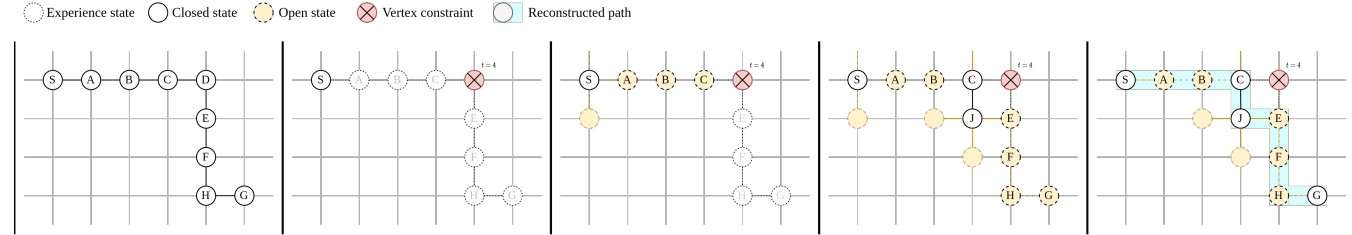

Figure 2: An illustration of our proposed algorithm accelerating a single agent search on a four-connected grid via reusing previous search efforts. **(a)** A single-agent path from $S$ to $G$ computed in a previous iteration. **(b)** Upon imposing a new constraint on the agent, shown in red, replanning is required. The previous path is drawn in light gray. **(c)** Upon expansion of node $S$, a prefix $\{A, B, C\}$ of the experience path is added to OPEN alongside all other successors of $S$. (c) shows two steps: node $C$ is selected for expansion from OPEN, and in the following iteration node $J$ is expanded from OPEN. Upon expanding $J$, a segment of the experience is added to OPEN since one of $J$'s successors is equivalent to a node in the experience. **(e)** Finally, $G$ is expanded from OPEN and the search terminates and recovers a path. In this example, the work done by xWA* (Alg. 2) is smaller than that of its previous iteration. By reusing experience, the intermediate nodes expanded are $C$ and $J$.

such as 2D-grid worlds, resulting in algorithms that often
65 rely on assumptions such as fast single-agent planning and
informative heuristics. These assumptions may not always
hold in other scenarios such as robotic manipulation. In this
study, we introduce a method aimed at speeding up multi-
agent path planning in contexts where single-agent planning
70 is hard.

## Planning for Multi-Arm Manipulation

In practice, planning for multi-arm manipulation is often
done with coupled methods or prioritization methods. In
coupled methods, the state of all arms is seen as a sin-
75 gle composite state, and the search is performed in this
space with algorithms such as A* (Nilsson 1980), Rapidly-
exploring Random Trees (RRT)(Karaman and Frazzoli
2011), and their variants (e.g., weighted A*, RRT* (Kara-
man and Frazzoli 2011), RRT-Connect (Kuffner and LaValle
80 2000), etc.). With the addition of more arms the search space
grows exponentially, and in general, coupled methods are
not scalable to large numbers of arms.

In scenarios where coupled planning is rendered in-
tractable due to the exponential growth of the search space,
85 prioritization methods may be effective in reducing its di-
mensionality. In prioritized planning (PP)(Erdmann and
Lozano-Perez 1987), each arm is assigned a priority, and
the lower-priority arms must respect the plans of all higher-
priority arms. In the general case, solving the prioritized
90 planning problem is more efficient than the coupled case,
as the search space is reduced to the space of each single
arm. However, the price paid for this dimensionality reduc-
tion is the loss of completeness. In scenarios requiring close
coordination between arms completeness may be important.
95 Recently, planning algorithms have been proposed for
teams of high-dimensional agents and applied to multi-arm
settings. These methods explore the search space via pre-
constructed single-agent roadmaps (probabilistic roadmaps
(PRM)(Kavraki et al. 1996) and potentially task-informed
100 roadmaps (Solano et al. 2023)), which may need to be ar-
bitrarily resampled (Solis et al. 2021) to find collision-free

paths in complex environments. In (Shome et al. 2020), the
authors present dRRT*, a method for exploring the com-
posite state-space of agents by traversing individual agents'
roadmaps towards sampled configurations with goal bias. In 105
(Solis et al. 2021), the authors present CBS-MP, a variant
of Conflict Based Search (CBS)(Sharon et al. 2015) that im-
poses new constraints on the search space to resolve con-
flicts between agents. Specifically, to resolve a conflict be-
tween two agents, CBS-MP requires one agent to avoid the 110
colliding configuration of the other at the time of conflict.

## Multi-Agent Path Finding

Multi-agent path finding (MAPF) is the problem of finding
collision-free paths for a set of agents on a graph (e.g., on
a grid world)(Stern 2019). MAPF has been studied exten- 115
sively, and optimal (e.g., CBS (Sharon et al. 2015)), bounded
sub-optimal (e.g., ECBS (Barer et al. 2014)), and sub-
optimal but effective (e.g., MAPF-LNS2 (Li et al. 2022))
algorithms have been proposed. Some work has also been
done to generalize MAPF algorithms to non-point robots 120
(Li et al. 2019), however, the most common domain is still
in 2D. Arguably, the most influential family of algorithms is
CBS and its extensions (Sharon et al. 2015; Barer et al. 2014;
Li, Ruml, and Koenig 2021). CBS is a two-level search al-
gorithm, where at the low level, each agent is assigned a 125
single-agent path planning problem. At the high level, con-
flicts between single-agent solutions are resolved by impos-
ing constraints on the low-level planners.

CBS is known to provide completeness and optimality
guarantees. However, CBS is also known to be computa- 130
tionally expensive as it requires repeated low-level searches
upon additions of constraints. Given this inefficiency, CBS
is often regarded as impractical for domains, like in manipu-
lation, where planning for a single agent requires the explo-
ration of a high-dimensional space and does not enjoy in- 135
formed heuristics. In this work, we capitalizing on this rep-
etition and propose a method for accelerating multi-agent
path finding algorithms by reusing online-generated previ-
ous search solutions.

## Leveraging Experience in Planning

Streamlining motion planning from experience encompasses a wide range of motion planning algorithms. These generally benefit from either utilizing offline-generated data (i.e., precomputation), from leveraging online-generated data, or both.

**Precomputation as Experience** The utilization of offline computations for enhancing online search efficiency is well exemplified by the PRM algorithm and its variants. Another novel approach is found in the Constant-time Motion Planners (CTMP) family of algorithms, which operates on precomputed data structures to achieve constant-time path generation in online scenarios (Islam, Salzman, and Likhachev 2021; Islam et al. 2021b,a; Mishani, Feddock, and Likhachev 2023). In recent research, a significant focus has been on the offline decomposition of the configuration space into collision-free convex sets (Dai et al. 2023). This decomposition enables planning smooth trajectories within these sets using optimization methods (Marcucci et al. 2023, 2022). Furthermore, various algorithms based on precomputed trajectories (Phillips et al. 2012; Berenson, Abbeel, and Goldberg 2012), have been employed to expedite the search process. When extending these techniques to plan for multi-arm setups, it becomes essential to decompose the composite configuration space for computing collision-free trajectories. However, challenges arise when the environment or the robot undergoes changes, which can be as simple as rotating a bin or altering the robot's geometry by grasping an item. These changes may require resource-intensive operations like redoing pre-computation or propagating changes, emphasizing a drawback inherent to using offline-generated experiences.

**Online-Generated Experiences** Anytime algorithms, like Anytime Repairing A* (ARA*) (Likhachev et al. 2008), can be seen as methods that utilize online-generated experiences to enhance solution quality over time. ARA* carries out a sequence of searches that, given enough time, converge to the optimal solution. A recent anytime approach inspired by (Likhachev et al. 2008; Phillips et al. 2012), and presented in (Mishani, Feddock, and Likhachev 2023), employs both precomputation and online experience. Their algorithm computes an initial, potentially sub-optimal, path within a (short) constant time and improves the quality of the path using the current best solution as an experience.

Drawing inspiration from the way the approach in (Mishani, Feddock, and Likhachev 2023) capitalizes on the flexibility seen in online-generated experiences, and with the observation that CBS-based algorithms inherently exhibit repetition in the form of nearly identical single-agent planning queries, we propose a method for accelerating multi-agent path finding algorithms by reusing online-generated experiences.

# Preliminary

In this paper, we propose a method for solving the M-RAMP problem by extending the CBS algorithm and its variants to reuse search efforts. We first describe the problem formulation and then detail the CBS algorithm.

## M-RAMP: Problem Formulation

Consider $\mathcal{Q}^i \subseteq \mathbb{R}^d$ as the configuration space of a single robotic arm $\mathcal{R}_i$ with $d$ degrees of freedom (DoF), and let the composite configuration space of $n$ robotic manipulators be $\mathcal{Q} = \mathcal{Q}^1 \times \mathcal{Q}^2 \times \cdots \mathcal{Q}^n$. With all manipulators operating within the same environment $\mathcal{W} \subset \mathbb{R}^3$, let $\mathcal{Q}_{\text{free}}$ be the set of all collision-free configurations (both with the environment and between robots):

$$\mathcal{Q}_{\text{free}} = \{q \in \mathcal{Q} \mid q \text{ is collision-free}\}$$

Given an initial composite configuration $q_{\text{start}} \in \mathcal{Q}_{\text{free}}$ and a composite goal configuration $q_{\text{goal}} \in \mathcal{Q}_{\text{free}}$, we want to find a valid path $\Pi : [0, T] \to \mathcal{Q}_{\text{free}}$ where $\Pi(0) = q_{\text{start}}$ and $\Pi(T) = q_{\text{goal}}$. A discrete analog of the problem is to find a sequence of configurations $\Pi = \{q_0, q_1, \cdots, q_T\}$ such that $\forall t \in [0, T]$, $q_t \in \mathcal{Q}_{\text{free}}$, each interpolated configuration between $q_t$ and $q_{t+1}$ is collision-free, and $q_0 = q_{\text{start}}$ and $q_T = q_{\text{goal}}$.

Instead of addressing the motion planning problem in the high-dimensional composite state space, it is possible to decompose the problem into a set of single-agent motion planning problems and locally resolve conflicts between the paths of agents. The resulting solution can be represented as $\Pi = \{\pi^1, \pi^2, \cdots, \pi^n\}$, where

$$\pi^i = \{q_0^i, q_1^i, \ldots, q_T^i \mid q_t^i \in \mathcal{Q}_{\text{free}}^i \quad \forall t = 0, \ldots, T\}$$

is a path for agent $\mathcal{R}_i$ from its start $q_{\text{start}}^i \in \mathcal{Q}^i$ to its goal $q_{\text{goal}}^i \in \mathcal{Q}^i$ configuration.

## Conflict Based Search

CBS is a complete and optimal two-level best-first search algorithm solving the MAPF problem. It utilizes single-agent planners, also known as low-level planners, to compute individual paths for each agent and employs a high-level search to resolve conflicts between the proposed paths.

CBS begins by querying a path $\pi_i$ for a given agent $\mathcal{R}_i$ between its start and goal configurations without regard to other agents. This solution $\Pi$ is a candidate solution for the problem, and it is stored in the OPEN list of the high-level search. A high-level node, called a constraint-tree (CT) node, holds within it a set of paths $\Pi$ for all agents, and a set of constraints $C$ imposed on the low-level planners. The cost of a CT node is the sum of the costs of its stored paths.

CBS proceeds iteratively, selecting least-cost solutions from OPEN and evaluating them for conflicts. If there are no conflicts found in a solution, then it is accepted as valid and the algorithm terminates. Otherwise, the conflict is used to create two new CT nodes, which are added to OPEN. Given a conflict between two agents $\mathcal{R}_i$ and $\mathcal{R}_j$ at time $t$, for example, because their configurations $q_t^i$ and $q_t^j$ are in collision, then two *vertex-constraints* are created. Either $\langle i, q_t^i, t \rangle$, forbidding $\mathcal{R}_i$ from being at $q_t^i$ at time $t$, or $\langle j, q_t^j, t \rangle$, forbidding $\mathcal{R}_j$ from being at $q_t^j$ at time $t$. If a conflict is found during a transition between times $t$ and $t + 1$, then *edge-constraints* are created and take the form $\langle i, q_t^i, q_{t+1}^i, t \rangle$ or

$\langle j, q_t^j, q_{t+1}^j, t \rangle$. Each edge-constraint forbids $\mathcal{R}_i$ or $\mathcal{R}_j$ from moving between $q_t^i$ and $q_{t+1}^i$ at time $t$.

Given the two new constraints created from the detected conflict, CBS and its variants create two new CT nodes. In each, the new constraint is added to the constraint set $C$, and the low-level planners are invoked to find a new path for each newly constrained agent. The new paths are stored in the new CT node, and the two created nodes are added to OPEN.

Enhanced CBS (ECBS) (Barer et al. 2014) is a widely used bounded sub-optimal variant that minimizes conflicts within a specified suboptimality bound. It employs focal-lists in low- and high-level searches, ordering nodes based on conflict minimization.

In CBS and its variants, new low-level planner invocations closely resemble previous ones. Initially invoked with constraints $C_i = \{c \in C \mid c \text{ involves } \mathcal{R}_i\}$ for agent $\mathcal{R}_i$, the next invocation includes $C_i \cup \{\langle i, q_t^i, t \rangle\}$ when an additional vertex constraint is introduced. This slight difference suggests potential benefits from reusing parts of the previous solution. Iterative-Deepening CBS (IDCBS)(Boyarski et al. 2021) leverages this insight in the 2D case using Lifelong Planning A* (LPA*)(Koenig, Likhachev, and Furcy 2004). However, this approach faces challenges in manipulation cases where bounded sub-optimal search is employed to navigate the high-dimensional search space (Likhachev and Koenig 2005).

## Algorithmic Approach

Our main contribution in this work is an experience-acceleration framework for CBS-based algorithms. We instantiate this framework in two incarnations, *xCBS* and *xECBS*, accelerating CBS and ECBS, respectively. In this section, we present the general form of our acceleration method in an intuitive manner grounded by Algorithm 1 and Algorithm 2, and then provide a theoretical analysis of its performance alongside its instantiations xCBS and xECBS.

### Experience-Acceleration Framework

Our framework follows the CBS structure and informs new low-level planner calls with the experience generated in previous search efforts. In the high-level search (Alg. 1), each node, called a CT node, contains a set of paths $\Pi$, one for each manipulator, and a set of constraints. Upon obtaining a new node from OPEN it is checked for conflicts (line 13). If there are none, the node is a goal node and the paths are returned (line 15). Otherwise, a set of constraints is derived from the conflicts (line 16). Usually, CBS proceeds by creating a new CT node, one with an added constraint from the constraint set (lines 18-19), and replans a single-agent path for each affected agent from scratch (line 21). However, we recognize that a considerable portion of the previously generated paths remains valid and can be effectively reused. Thus, to speed up the search, we cache a copy of the previously computed paths as *experience*, which are in turn passed to the low-level motion planner (lines 17, 21). The experiences are time-agnostic, meaning that they do not include a time dimension but only the topology of the path. We

---

**Algorithm 1:** High-level Planner

**Input** : $n$: Number of manipulators (agents)
$\quad q_{\text{start}} = \{q_{\text{start}}^0, \cdots q_{\text{start}}^n\}$
$\quad q_{\text{goal}} = \{q_{\text{goal}}^0, \cdots q_{\text{goal}}^n\}$

**Output:** Path $\Pi = \{\pi^1, \pi^2, \cdots, \pi^n\}$ from start to goal states

1 **Procedure** InitRootNode()
2 $\quad$ RootNode.constraints $\leftarrow \emptyset$
3 $\quad$ RootNode.paths $\leftarrow$ invoke Planner for each agent
4 $\quad$ RootNode.cost $\leftarrow$ GetCost (RootNode.paths)
5 $\quad$ **return** RootNode

6 **Procedure** Plan ($n$, $q_{start}$, $q_{goal}$)
7 $\quad$ RootNode $\leftarrow$ InitRootNode()
8 $\quad$ Insert RootNode to OPEN
9 $\quad$ **while** *OPEN not empty* **do**
10 $\quad\quad$ FOCAL $\leftarrow \{n | f_1(n) \cdot \leq w \cdot \min_{n' \in \text{OPEN}} f_1(n')\}$
11 $\quad\quad$ Node $= \underset{n \in \text{FOCAL}}{\arg\min} f_2(n)$
12 $\quad\quad$ OPEN.pop(Node)
13 $\quad\quad$ conflicts $\leftarrow$ FindConflicts (Node.paths)
14 $\quad\quad$ **if** *conflicts* $= \emptyset$ **then**
15 $\quad\quad\quad$ **return** Node.paths
16 $\quad\quad$ constraints $\leftarrow$ GetConstraints (conflicts.first)
17 $\quad\quad$ *Experiences* $\leftarrow$ RemoveTime (Node.paths)
$\quad\quad$ // Removing time from path states, so we could use them as experiences
18 $\quad\quad$ **for** $c \in$ *constraints* **do**
19 $\quad\quad\quad$ Create new CT node *NewNode*
20 $\quad\quad\quad$ *NewNode*.constraints $\leftarrow$ Node.constraints $\cup$ c
21 $\quad\quad\quad$ *NewNode*.paths $\leftarrow$ {
$\quad\quad\quad\quad$ Planner.Solve (*Experiences*[i], $q_{start}^i$, $q_{goal}^i$)
$\quad\quad\quad\quad$ if $i \in c$ else Node.paths[i]
$\quad\quad\quad\quad$ } // Invoke Planner for each agent involved
22 $\quad\quad\quad$ *NewNode*.cost $\leftarrow$ GetCost (*succ*.paths)
23 $\quad\quad\quad$ OPEN.insert(*NewNode*)
24 $\quad$ **return** $\emptyset$

---

have experimented with reusing experiences from the previous search effort, from all previous search efforts on the CT branch, and globally from the CT, and seen that reusing the previous path yields the best performance.

The low-level of our acceleration framework, namely $xWA^*$, is detailed in Algorithm 2 and illustrated in Fig. 2. Each node expansion (lines 17-32) adds a set of successors to the OPEN list. Additionally, upon a node expansion, xWA* attempts to accelerate the search by also adding a subset of the experience path to the OPEN list.

Upon a choice of a node for expansion (line 17), the search terminates if it is a goal state (lines 18-21). Otherwise, we check if the expanded state belongs to the experience path (line 23). The experience and the goal are strictly spatial, so state equivalence does not include time. If the expanded state belongs to the experience, and at least one consecutive state in the experience is feasible, we say that the state satisfies the *addition-condition*. Subsequently, starting from that state, we aim to add as much of the experience

as possible to the OPEN list (line 25). This process is also applied to the start state (line 14) and essentially provides a "warm start" to the search effort.

Given an expanded state $s$ that satisfies the *addition-condition*, we first propagate the time and cost of the experience to begin at the values of $s$ (line 3). Then, we attempt to add consecutive states from the experience (line 5) until we encounter the *termination-condition*, namely, violating constraints.

The effect of adding an experience path to the OPEN list of a bounded sub-optimal search algorithm, such as weighted A*, could be a rapid exploration of states that are closer to the end of the experience path (and consequently, closer to the goal). Figure 2 illustrates this effect. Such exploration results in the algorithm "jumping" over previously explored regions and avoiding redundant search efforts, directing its focus closer to the end of the experience.

Collision checking against the static environment, a significant factor in the slowness of planning for manipulation, can also be directly accelerated with experience. To this end, our acceleration framework also keeps track of the configurations $(q_t^i, q_{t+1}^i)$ in all valid transitions $(s_t, s_{t+1})$ for each robot $\mathcal{R}_i$. With this information, the successors set (line 26) can be computed more rapidly by only checking the validity of edges previously unseen. Since it is possible for one single-agent search to revisit the same configuration at different times, such experience reuse also speeds up the first search.

## Theoretical Analysis

In this section, we discuss the theoretical foundation of our algorithm. We show that it is complete and provide bounded sub-optimality guarantees. First, we define the CBS framework using *focal-search* and introduce some of the properties of CBS and its bounded sub-optimal variants. Subsequently, we demonstrate that our accelerated variant maintains these properties.

We formally define the problem for both levels of CBS as a *focal-search* (Cohen et al. 2018). Focal search employs two priority queues: OPEN and FOCAL. OPEN mirrors the A* queue using $f_1$ as its priority function, while FOCAL comprises a subset of OPEN defined as FOCAL $= \{n | f_1(n) \le w \cdot f_{1_{min}}\}$, where $w$ denotes the sub-optimality factor. Then, FOCAL utilizes the priority function $f_2$ to order its nodes. Assuming the admissibility of $f_1$, we are guaranteed that the returned solution is at most $wC^*$, where $C^*$ is the cost of the optimal solution (Pearl and Kim 1982). Consequently, to reason about the total sub-optimality bound of CBS variants, it would suffice to formulate their high- and low-level planners as instances of focal search each contributing a factor to the total sub-optimality bound. In the following paragraphs, we define the sub-optimality factor *contribution* by a focal search, detail the total sub-optimality bound of a two-level focal search, and finally show the completeness and sub-optimality bounds of xCBS and xECBS.

**Definition 1.** *We say that a sub-optimality factor **contributed** by a focal-search with admissible $f_1$ function is a*

---

**Algorithm 2:** xWA*: Low-level Planner

**Input** : $q_{start}$: start state ($q_{start} \in \mathcal{Q}^{free}$)
$q_{goal}$: goal state ($q_{goal} \in \mathcal{Q}^{free}$)
$\tilde{\pi}$: Experience path (without time)
$w_1, w_2$: sub-optimality bounds for WA* and focal list.

**Output:** Path $\pi$

1 **Procedure** PushPartialExperience($\tilde{\pi}$, *OPEN*, $s$)
2  $\quad \hat{\pi} \leftarrow \tilde{\pi}$.suffix($s$);    // Slicing the experience to extract all states beginning from $s$.
3  $\quad \hat{\pi} \leftarrow$ PropagteTimeAndCost ($\hat{\pi}$, $s$.time, $s.g$)
4  $\quad$ **for** $s_t \in \hat{\pi}$ **do**
5  $\quad\quad$ **if** IsEdgeValid ($s, s_t$) $\land$ IsStateValid ($s_t$) **then**
6  $\quad\quad\quad$ insert $s_t$ to OPEN
7  $\quad\quad\quad$ $s \leftarrow s_t$
8  $\quad\quad$ **else**
9  $\quad\quad\quad$ break

10 **Procedure** Solve($q_{start}$, $q_{goal}$, $\tilde{\pi}$)
11  $\quad \pi = \emptyset$ ; CLOSED $= \emptyset$ ; FOCAL $= \emptyset$ ;
$\quad f_1(s) := g(s) + w_1 h(s)$
12  $\quad s \leftarrow (q_{start}, 0)$;   // Adding time to state.
13  $\quad$ OPEN $= \{s\}$ ; remove $q_{start}$ from $\tilde{\pi}$
14  $\quad$ PushPartialExperience($\tilde{\pi}$, *OPEN*, $s$)
15  $\quad$ **while** *OPEN* $\ne \emptyset$ **do**
16  $\quad\quad$ FOCAL $\leftarrow \{s | f_1(s) \le w_2 \min_{s' \in \text{OPEN}} f_1(s')\}$
17  $\quad\quad$ $s_{min} = \underset{s \in \text{FOCAL}}{\arg\min} f_2(s)$
18  $\quad\quad$ **if** IsGoalCondition($s_{min}$) // The state is at $q_{goal}$ and there are no future constraints.
19  $\quad\quad$ **then**
20  $\quad\quad\quad$ $\pi \leftarrow$ ExtractPath()
21  $\quad\quad\quad$ break
22  $\quad\quad$ insert $s_{min}$ into CLOSED
23  $\quad\quad$ **if** RemoveTime($s_{min}$) $\in \tilde{\pi}$ **then**
24  $\quad\quad\quad$ $s = s_{min}$
25  $\quad\quad\quad$ PushPartialExperience($\tilde{\pi}$, *OPEN*, $s$)
26  $\quad\quad$ **for** $s' \in Successors(s_{min})$ **do**
27  $\quad\quad\quad$ **if** $s'$ *was not visited before* **then**
28  $\quad\quad\quad\quad$ $g(s') = \infty$
29  $\quad\quad\quad$ **if** $g(s') > g(s_{min}) + c(s, s')$ **then**
30  $\quad\quad\quad\quad$ $g(s') = g(s_{min}) + c(s, s')$
31  $\quad\quad\quad\quad$ **if** $s' \notin CLOSED$ **then**
32  $\quad\quad\quad\quad\quad$ insert $s'$ into OPEN
33  $\quad$ **return** $\pi$

---

*constant $w$, such that for every expanded node $N$:*

$$f_1(N) \le wC^*$$

Let us first define the high-level search as focal search with $f_1 = f_1^H = g(n)$, where $g(n)$ is the cost of the CT node (sum of agents' path costs), and $f_2 = f_2^H$ to be some priority function. Such a focal search contributes a given sub-optimality factor $w_H$.

**Lemma 1.** *Let $w_H, w_L$ be the sub-optimality factor contributed by the high- and low-level focal searches, respectively. For any $w_H, w_L \ge 1$, the cost of the solution is at most $w_H w_L C^*$.*

*Proof.* Let $N$ be a node in FOCAL of the high-level search. Additionally, Let $k$ be the number of agents (manipulators) each having a returned *cost*. We denote the returned cost of the $i^{\text{th}}$ agent low-level planner as $cost(i)$ and its optimal cost as $cost^*(i)$. Lastly, let $f_{1,i}^L(s|n)$ be the $i^{\text{th}}$ agent low-level planner's priority function, within a given high-level node $n$, and let $s_{g,i}$ be the goal state for agent $i$.

$$N.cost = g(N) = f_1^H(N) \leq w_H \min_{n \in OPEN} f_1^H(n)$$

$$= w_H \min_{n \in OPEN} \sum_{i=1}^k cost(i) = w_H \min_{n \in OPEN} \sum_{i=1}^k f_{1,i}^L(s_{g,i}|n)$$

$$\leq w_H \sum_{i=1}^k w_L cost^*(i) = w_H w_L C^*$$

$\square$

Lemma 1 implies that proving the bounded sub-optimality of our approach necessitates showing that both the high-level search and the low-level search are focal searches each contributing a sub-optimality factor.

We commence by establishing the bounded sub-optimality of the low-level planner xWA*, which leverages past experiences and contributes a factor of $w_L$. To allow for the use of inflated heuristics using a weighted OPEN (Veerapaneni, Kusnur, and Likhachev 2023) list, which is common in manipulation, we expand our analysis to low-level planners with $w_1$-*admissible* (Pearl and Kim 1982) priority function $f_1$.

**Lemma 2.** *Considering a focal search that employs a $w_1$-admissible function $f_1(s)$ ($w_1 \geq 1$) and FOCAL= $\{s | f_1(s) \leq w_2 \min_{s' \in OPEN} f_1(s')\}$, the contributed sub-optimality factor is $w_1 \cdot w_2$.*

*Proof.* Let $n_0$ be a node on an optimal path which resides in OPEN. For every expanded node $N$:

$$f_1(N) \leq w_2 \min_{n \in OPEN} f_1(n) \leq w_2 f_1(n_0) =$$

$$= w_2(g(n_0)+w_1 h(n_0) \leq w_2 w_1(g(n_0)+h(n_0) \leq w_2 w_1 C^*$$

$\square$

Hence, our remaining task is to show that incorporating experiences in xWA* does not impact the contributed sub-optimality factor, nor sacrifices completeness.

**Lemma 3.** *Consider a best-first search storing frontier states in an OPEN list. When systematically incorporating successors into OPEN, if additional nodes are introduced along with their associated $f$ values, the properties of completeness and bounded sub-optimality persist.*

*Proof.* As we introduce new nodes to OPEN, the original OPEN of weighted A* becomes a subset of the modified OPEN. Moreover, the algorithm maintains its systematic nature, ensuring completeness. Furthermore, we also know that FOCAL will only be populated by nodes from OPEN that are within the specified sub-optimality bound. Consequently, when a goal state is expanded, the solution remains bounded sub-optimal. $\square$

**Theorem 1.** *xWA* is complete and bounded sub-optimal, contributing $w_L = w_1 w_2$.*

*Proof.* Since xWA* is a focal search, which employs a weighted OPEN ($w_1$-admissible $f_1$), the proof follows directly from Lemma 2 and 3 $\square$

We initially assumed that the high-level search is given and that it is bounded sub-optimal contributing factor of $w_H$. In what follows we will discuss the conditions under which the high-level search algorithm presented in Alg. 1 is complete and bounded sub-optimal.

To show the completeness of the high-level search of CBS-variants, we turn our attention to the way they impose constraints on the low-level searches. A CBS-variant's high-level search is complete if, when it creates constraints $c_1$ and $c_2$ for resolving a conflict, then there exists no valid solution that invalidates $c_1$ and $c_2$. Otherwise, valid solutions with respect to conflicts will be marked as invalid with respect to constraints. Interestingly, by viewing the high-level completeness of CBS variants in this way, it can be shown that some CBS variants, such as CBS-MP, gain efficiency by sacrificing completeness despite initially claiming otherwise[1].

**Theorem 2.** *Our proposed acceleration framework is complete and bounded sub-optimal.*

*Proof.* Assuming the use of valid constraints, the focal list on the high-level is complete and bounded sub-optimal with a factor of $w_H$, as shown in (Barer et al. 2014). From Theorem 1, we have that xWA* is complete and bounded sub-optimal by $w_L$. Therefore, Lemma 1 shows a sub-optimality upper bound of $w_H w_L C^*$ for xCBS. $\square$

Under this structure, we will show that CBS and ECBS are complete and (bounded sub-) optimal and show that xCBS and xECBS are complete and bounded sub-optimal.

**CBS** the suboptimality factor contributed by the low-level search is $w_L = 1$ since it is usually an optimal search (e.g., A*), and the high-level search does not employ a focal-list and prioritizes CT nodes according to their sum-of-costs. CBS is complete since it imposes constraints only on the vertex or edge that was in conflict (Sharon et al. 2015); invalidating both constraints leads back to the conflicting configuration found in the first place. Thus, we restate that CBS is complete and optimal.

**ECBS** the suboptimality factor contributed by the low-level search is a user-defined constant $w_L$, implemented as a focal-list. In the high-level, an adaptive focal list steers the search but does not contribute an additional sub-optimality factor (i.e., $w_H = 1$) due to its dependence on the lower bound of the low-level searches (Barer et al. 2014). ECBS is complete since it imposes similar constraints to CBS.

**xCBS** At the high-level, xCBS is identical to CBS in both its CT node prioritization (according to their sum-of-costs) and its constraint generation function. Thus, it contributes a $w_H = 1$ and maintains completeness. At the low level, our

---

[1]We have discussed CBS-MP's theoretical guarantees with the authors and reached this conclusion.

xWA* is complete and contributes a sub-optimality factor of $w_L$, as shown in Theorem 1.

**xECBS** At the low level, xWA* contributes a sub-optimality factor of $w_L = w_1 w_2$. At the high-level, the contributed sub-optimality factor is $w_H = 1$ owing to the adaptive bound used in ECBS. Completeness is guaranteed for the same reasons as xCBS. xECBS terminates the addition of experiences at detected collisions with other agents traveling on their previously computed paths. Since ECBS prioritizes low-level search states based on their added conflicts, we refrain from creating new conflicts when reusing experiences.

In this light, we have shown that xCBS and xECBS maintain completeness and bounded sub-optimality guarantees while also being accelerated.

## Experiments

To evaluate xECBS and xCBS, we created collaborative manipulation tasks with varying numbers of robots, obstacle density, and robot-robot interaction complexity. Each robot in our experiments is a Franka Panda manipulator with 7-DOF. The experiments were conducted on an Intel Core i9-12900H with 32GB RAM (5.2GHz).

### Experiments Setup

Our experiments focus on testing the scalability of algorithms as well as their applicability for real-world use. We set up 7 scenes, each with 50 planning problems defined by starts $q_{\text{start}} \in \mathcal{Q}_{\text{free}}$ and goals $q_{\text{goal}} \in \mathcal{Q}_{\text{free}}$.

To test the applicability of algorithms for real-world scenarios, we evaluated algorithms in two sample tasks: shelf rearrangement with 8 arms and bin-picking with 4 arms. For each scene, we randomly sampled 50 start and goal states from a set of task-specific configurations (e.g., pick and place configurations at different bins or positions in between shelves). Given that the robots operate within the same task-space, these configurations require motion plans with substantial proximity between arms.

To shed light on how algorithms scale with the number of arms, we tested their performance in free or lightly cluttered scenes with 2, 4, 6, 8, and 10 arms as shown in Figure 4. The starts and goals for each agent are in the shared workspace region. In each setup, robots were organized in a circle, and in the cases with 6, 8, and 10 robots, a thin obstacle was placed in the circle to encourage interaction.

### Baselines

To show the efficacy of our method, we compare it both to ubiquitous algorithms commonly used to solve the M-RAMP problem, as well as to other algorithms recently applied to M-RAMP.

**Sampling-Based Methods** We include PRM and RRT-Connect, which are arguably the most commonly used algorithms for planning in manipulation. For both, the search space is the composite state space $\mathcal{Q}$. We use the OMPL (Sucan, Moll, and Kavraki 2012) implementation of PRM and RRT-Connect. Additionally, we include dRRT*(Shome

et al. 2020), a more recent algorithm applied to M-RAMP that explores the composite state space via transitions on single-agent roadmaps. In our implementation, the single-agent roadmaps contain a minimum of 1500 nodes, with increments of 1000 more being sampled if the roadmap cannot be connected to the start or goal configurations.

**Search-Based Methods** We include PP, CBS, ECBS, and CBS-MP in our comparison, as well as our proposed methods xCBS and xECBS. For all, the single agent planners are weighted A* with a uniform cost for transition and an $L_2$ joint-angle distance as a heuristic. The heuristic inflation value is 50 and in ECBS and xECBS the sub-optimality bound is set to 1.3. Our implementation of CBS-MP differs slightly from the original in that, here, agents plan on uniformly discretized implicit graphs and not on precomputed roadmaps. This has been done to compare all search algorithms on the same planning representation. All edge transitions on the implicit graphs are said to take one timestep.

### Evaluation Metrics and Postprocessing

We are interested in the scalability and solution quality of algorithms. To this end, for each scene, we report the mean and standard deviation for planning time and solution cost across all segments, alongside the success rate of each algorithm in the scene. All algorithms were allocated 60 seconds for planning, after which a plan was considered a failure. The cost is the total motion carried out by the joints, in radians. In our scalability analysis, we also add metrics for the number of collision checks carried out by a subset of the algorithms.

All solutions are post-processed with a single pass of a simple incremental shortcutting algorithm. One by one, each agent's solution path is shortcutted without creating new conflicts. Starting from the beginning of the path, the algorithm attempts to replace path segments by linear interpolations while avoiding obstacles and other agents. This standard shortcutting algorithm is often used to refine paths yielded by sampling-based planners.

### Experimental Results

We observe that xECBS solves real-world multi-arm manipulation planning problems faster and with a higher success rate compared to other evaluated methods while keeping solution costs low. Figure 3 (middle) illustrates this result. The figure shows the pairwise relative cost and runtime of all successful algorithms, where the values are computed over jointly successful problems. xECBS demonstrates faster planning speed (values above 100%) while delivering low-cost solutions comparable to those achieved by other conflict-based approaches (values around to 100%). Comprehensive statistics for all runs are provided in the accompanying tables.

Our scalability analysis shows that xECBS scales to scenes with many agents better than competing methods, consistently finding solutions for more problems. We note that xCBS improves on CBS in general, however, xECBS offers a much larger boost in performance and is more suitable for planning for multi-arm manipulation.

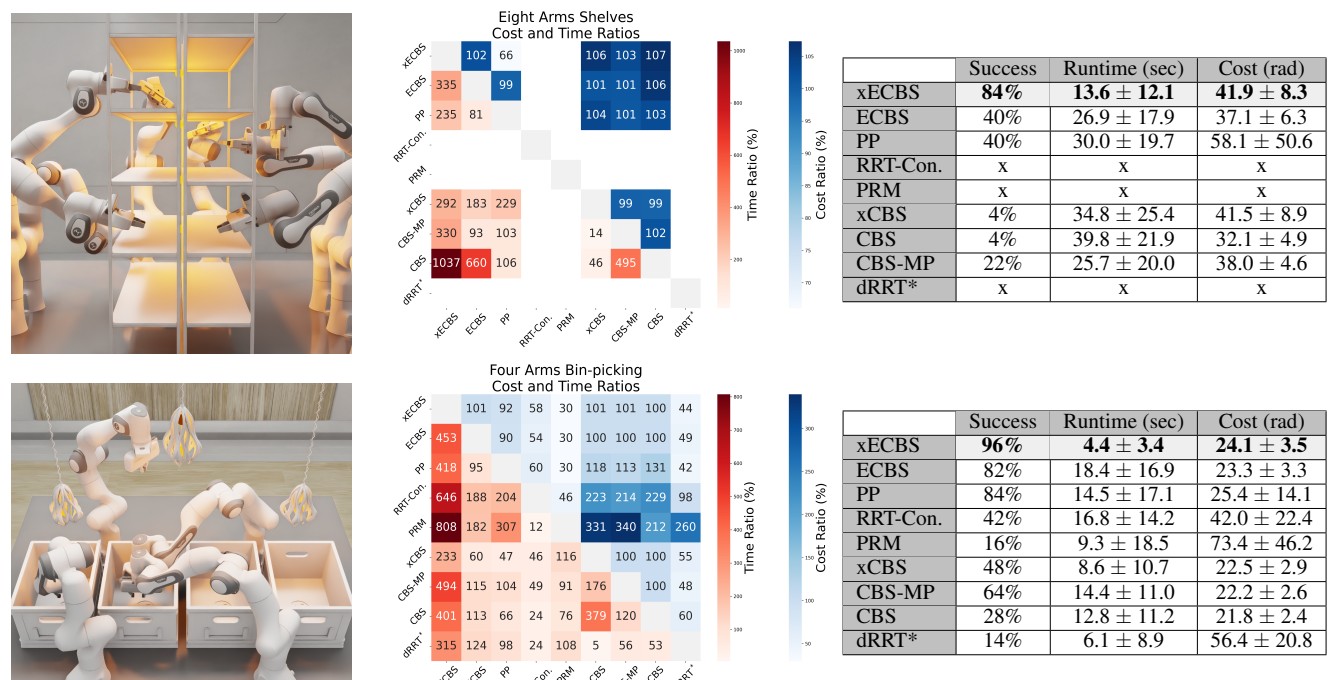

Figure 3: Evaluating the real-world applicability of planning algorithms. **Left**: evaluation scenes, with 8-arm shelf rearrangement and 4-arm bin-picking. **Middle**: Comparing runtime and cost ratios between methods. Values are the ratio (vertical to horizontal) between methods averages in jointly solved problems. xECBS is faster and finds short paths. **Right**: Statistics.

Eight Arms Shelves:

| | Success | Runtime (sec) | Cost (rad) |
|---|---|---|---|
| xECBS | **84%** | **13.6 ± 12.1** | **41.9 ± 8.3** |
| ECBS | 40% | 26.9 ± 17.9 | 37.1 ± 6.3 |
| PP | 40% | 30.0 ± 19.7 | 58.1 ± 50.6 |
| RRT-Con. | x | x | x |
| PRM | x | x | x |
| xCBS | 4% | 34.8 ± 25.4 | 41.5 ± 8.9 |
| CBS | 4% | 39.8 ± 21.9 | 32.1 ± 4.9 |
| CBS-MP | 22% | 25.7 ± 20.0 | 38.0 ± 4.6 |
| dRRT* | x | x | x |

Four Arms Bin-picking:

| | Success | Runtime (sec) | Cost (rad) |
|---|---|---|---|
| xECBS | **96%** | **4.4 ± 3.4** | **24.1 ± 3.5** |
| ECBS | 82% | 18.4 ± 16.9 | 23.3 ± 3.3 |
| PP | 84% | 14.5 ± 17.1 | 25.4 ± 14.1 |
| RRT-Con. | 42% | 16.8 ± 14.2 | 42.0 ± 22.4 |
| PRM | 16% | 9.3 ± 18.5 | 73.4 ± 46.2 |
| xCBS | 48% | 8.6 ± 10.7 | 22.5 ± 2.9 |
| CBS-MP | 64% | 14.4 ± 11.0 | 22.2 ± 2.6 |
| CBS | 28% | 12.8 ± 11.2 | 21.8 ± 2.4 |
| dRRT* | 14% | 6.1 ± 8.9 | 56.4 ± 20.8 |

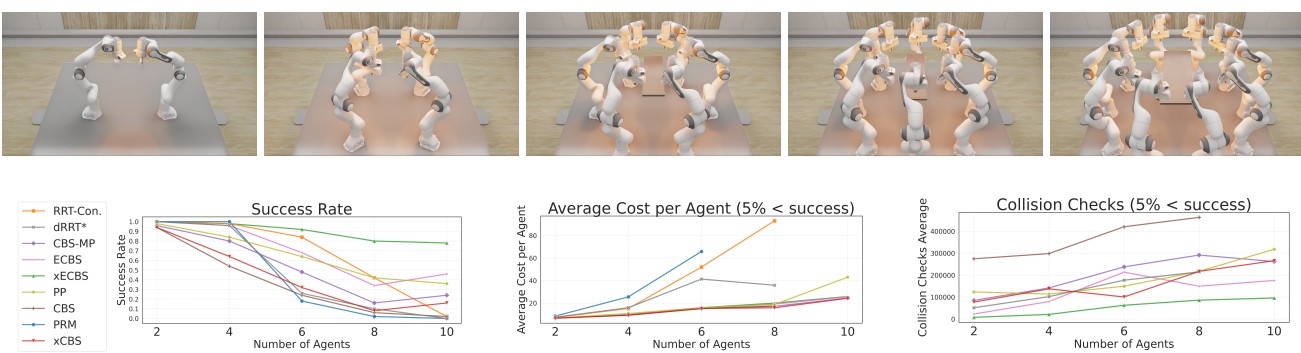

Figure 4: Scalability analysis. **Top**: an illustration of our test scenes with 2, 4, 6, 8, and 10 robots. **Bottom left**: the success rate of methods among the 50 planning problems in each scene. xECBS scales better than the competing method. **Bottom middle**: the average cost per robot of successful runs. We observe that all conflict-based methods maintain similar costs while PP and sampling-based methods produce worse paths even after shortcutting. **Bottom right**: the number of collision checks carried out on average by each one of the methods.

## Conclusion

Popular multi-agent motion planning algorithms like CBS and ECBS assume fast single-agent planners, which may not be available in multi-arm manipulation tasks. To address this, we propose to accelerate conflict-based algorithms by reusing online-generated path experiences and demonstrate their benefits in xCBS and xECBS. These adaptations improve performance in multi-arm manipulation tasks while ensuring bounded sub-optimality guarantees. Our experiments demonstrate the proposed method's effectiveness in various multi-arm manipulation tasks with up to 10 arms. We observe that xECBS is particularly effective in real-world scenarios such as pick and place and shelf rearrangement, achieving higher success rates and lower planning times than currently available methods.

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
