# OpenReview forum: "Accelerating Search-Based Planning for Multi-Robot Manipulation by Leveraging Online-Generated Experiences"
_icaps-conference.org/ICAPS/2024/Conference — ICAPS 2024_

### Official Review · Reviewer_q8Pt · 2024-01-13

**Significance And Importance:** 3
**Soundness:** 3
**Novelty:** 3
**Clarity:** 3
**Overall Evaluation:** 3
**Confidence:** 4

**Weaknesses:**

1: Minor weaknesses that are easily fixable.

**Contributions Of The Paper:**

This paper proposes an approach for accelerating conflict-based search algorithms by reusing past search results to solve the problem of multi-arm manipulation. The main contributions of this paper are 1) an enhanced conflict-based search method for multi-robot-arm motion planning, 2) theoretical analysis on sub-optimality and completeness, and 3) empirical evaluations.

**Ethical Considerations:**

(1) Not Applicable: The paper does not have any ethical considerations to address

**Nomination For Best Paper:**

Yes

**Questions For Authors:**

Major:
•	In Experimental Results,
-	It is not clear to me what are time ratio and cost ratio. Please provide definitions.
-	It is not clear to me how to read the middle part of Figure 3. Please elaborate.
•	For self-rearrangement, please elaborate why
-	xCBS has such a low success rate;
-	xCBS has a higher time consumption than most other methods;
-	xECBS has a higher cost than most other methods;
•	For bin-picking, please elaborate why
-	xCBS has a low success rate;
-	xCBS has higher time consumption than dRRT*;
-	xECBS has a higher cost than some algorithms.

Manor:
•	Figure 1 is not ever referenced.
•	In Figure 2, ‘(c) shows two steps: …’ it seems the two steps are shown by (c) and (d),
•	In Algorithmic Approach Line 335, Figure 2 is placed too far.
•	In Experiments Line 525, Figure 4 is referenced before Figure 3. Could you rearrange the figures such that the experiment scenarios are introduced first.

**Reproducibility:**

3: Authors describe the implementation and domains in sufficient detail.

**Strengths Of The Paper:**

•	The paper is well written,
•	The general idea is interesting and feasible,
•	The experiments are solid.

**Weaknesses Of The Paper:**

•	Experiments results not clearly explained,
•	Misplaced figures.

---

> ### Author Rebuttal · Authors · 2024-01-28
>
> Dear Reviewer q8Pt, thank you for your valuable feedback. We are committed to addressing your concerns to improve our paper's clarity and robustness.
>
> **Experimental Results**:
> The table in Fig.3 (middle) is a pairwise comparison between planners, showing the ratio (%) of planning time or plan cost for each row-name planner relative to the column-name planner. The values offer a "fair" comparison by considering only successful runs in both planners. For instance, xECBS has shorter planning times (red, above 100%) and lower solution costs comparable to other conflict-based approaches (blue, around 100%, lines 581-584 of the manuscript). We will add details to the caption to aid with understanding the comparison table.
>
> * **Shelf Rearrangement**
>
> 1. _xCBS success rate_:
> The success rate of xCBS is reduced as it inherits some inefficiencies from CBS, which also struggles in this experimental setup. Neither CBS nor xCBS employs conflict-based heuristics. Consequently, despite the acceleration of the low-level planner, xCBS may still need to explore a substantial portion of the CT, leading to time-consuming operations.
>
> 2. _xCBS time consumption_:
> Given that the success rate of xCBS was only 4%, we believe it may not be representative enough to show a trend. However, in the bin-picking setup, where the success rate is higher, the comparison matrix suggests that xCBS tends to find solutions faster than most other methods.
>
> 3. _xECBS cost_:
> The xECBS cost in the right table is averaged over a larger number of successful experiments and therefore is higher. Yet, the comparison matrix shows its cost is on par with other conflict-based variants and lower than PP.
>
> * **Bin-Picking**
>
> 1. _xCBS success rate_:
> Same reasons as above, however, here, we see that xCBS is significantly better than CBS.
>
> 2. _xCBS time consumption vs. dRRT*_:
> Here, dRRT* solved only 14% of the experiments, which we hypothesize were “simpler.” Being a sampling-based approach, dRRT* may find solutions faster in simpler cases compared to heuristic search-based approaches that systematically explore the search space.
>
> **Figure 1**: Thank you. We will add a reference to the figure in the introduction.
>
> **Figure 2 Correction**: Thank you for catching this typo. We will correct (c) to (d).
>
> **Figure 2 Placement**: Thank you. We will move the figure to the page where it is first referenced.
>
> **Figure 4 Referenced Early**: We will add an earlier reference to Figure 3.
>
> Sincerely,
>
> The Authors

---

### Official Review · Reviewer_S3Jg · 2024-01-23

**Significance And Importance:** 2
**Soundness:** 3
**Novelty:** 3
**Clarity:** 3
**Overall Evaluation:** 1
**Confidence:** 3

**Weaknesses:**

1: Minor weaknesses that are easily fixable.

**Contributions Of The Paper:**

This paper explores the problem of robot manipulation using multiple robot arms (M-RAMP - Multi-Robot-Arm Motion Planning). Building on the use of conflict-based methods for multi-agent path finding (MAPF), the paper proposes a new method for accelerating motion planning by making use of the repetitive and incremental nature of such algorithms to leverage experiences gathered during the planning process. In particular, online-generated path experiences are used to speed up multi-agent search, while preserving completeness and suboptimality guarantees. Building on Conflict-Based Search (CBS) and Enhanced CBS (ECBS), the paper proposes new high-level and low-level planning algorithms, that incorporate the experience-acceleration framework. These extensions are compatible with the existing (E)CBS algorithms, resulting in accelerated versions, xCBS and xECBS.  Theoretical results are established for the new algorithms, establishing that they are complete and bounded sub-optimally. In addition, an empirical evaluation is performed comparing standard methods (PP, CBS, ECBS, CBS-MP) against the accelerated version on a set of collaborative manipulation tasks, with xECBS showing high success rates and lower runtimes compared with the baselines.

**Ethical Considerations:**

(1) Not Applicable: The paper does not have any ethical considerations to address

**Nomination For Best Paper:**

No

**Questions For Authors:**

1) Perhaps the authors could comment on which parts of the original CBS/ECBS algorithms are kept and what their new algorithms replace/change (possibly referring to lines in the provided algorithms)? This wasn't completely clear in the text.

2) Are the experiments performed in simulation or using real robots (or a combination of both)?

3) Were there particular tasks that were particularly challenging for the new algorithms? It's difficult to tell from the combined results. Perhaps the authors could comment on this point.

Post-rebuttal comments: I thank the authors for their answers to my questions, which have been very helpful. I stand by my comments and score.

**Reproducibility:**

2: Some details are missing, but the paper still appears to be replicable with some effort.

**Strengths Of The Paper:**

This paper has several strengths. M-RAMP is an interesting and challenging problem and one that has practical applications across a range of real-world collaborative tasks. The contributions in the paper are both algorithmic and theoretical, supported by an empirical evaluation: new algorithms for the different levels of search are provided, and the properties of these algorithms are formally analysed. The empirical evaluation provides good evidence to support the efficiency claims in the paper, by comparing against a set of standard methods. (The video in the supplemental material provides some nice examples of the algorithm being applied in practice.) The paper is also generally well written and understandable.

**Weaknesses Of The Paper:**

This paper has several weaknesses:

- It would be useful to include a bit more intuition in the M-RAMP problem formulation section of the paper, especially for readers who are not experts in this area. The focus on different types of search in the paper should make it accessible to a wide range of planning researchers, but making the ideas in the formal description of the problem clear would be helpful (e.g., properties of the environment, the notion of configurations).

- The notion of conflicts in the CBS description is only briefly mentioned. I wasn't sure if these were simply collision points or if there was a more general notion of collision being used (e.g., overlapping 3D regions, points that are close but not strictly speaking in collision). Perhaps this could be discussed a bit more in the text.

- Conflict based search is described in the paper but the original CBS and ECBS algorithms are not provided. The text seems to suggest that the new high-level and low-level algorithms only replace parts of the original CBS/ECBS algorithms, but this isn't completely clear. It would be useful to more directly indicate what is changed with these algorithms and what is retained from the original CBS/ECBS algorithms.

- In the high-level planning algorithm, when the "Planner" is invoked for each agent in line 3, it wasn't completely clear what planner is being used here. Line 21 seems to invoke the low-level planner but line 3 wasn't clear.

- In the description of the experiments, it would be good to clarify whether they were done in simulation or using real robots (or possibly a combination of both). If they were done in simulation, it would be interesting to know what software was used.

- For the empirical results, it might be useful to break down the planning times for the different experiments, to illustrate the scalability of the approaches. Right now, the planning times seem to be combined in a single runtime value in the tables.

---

> ### Author Rebuttal · Authors · 2024-01-28
>
> Dear Reviewer S3Jg, thank you for your valuable feedback. We are committed to addressing your concerns to improve our paper's clarity and robustness.
>
> **Problem Formulation**: Thank you for your suggestion to provide more intuition. We will modify the Problem Formulation section and expand on the notion of a configuration (joint angles), and the challenge of planning amidst obstacles in manipulation tasks, which involves determining arm occupancy via forward kinematics and collision checking.
>
> **Defining Conflicts**: We follow a very similar conflict definition to the one presented in the CBS paper. To improve clarity, we will modify the Conflict Based Search section to include that two agents are said to be in conflict at some time if their configurations at that time are in collision.
>
> **Algorithm Changes**: We appreciate your suggestion to clarify differences with (E)CBS. We will modify lines 298-300 to underscore that the high-level planner has minimal changes (Alg. 1 lines 17, 21), and clarify in lines 307-310 that the low-level planner differs from the standard best-first structure mainly in its ability to detect applicable experiences for expanded states and accelerate the search with valid sub experiences.
>
> **Algorithm Text**: Thank you for highlighting the ambiguity with "planner." We'll update the algorithm to use LLPlanner and HLPlanner as distinct names.
>
> **Experiments**: Our experiments were conducted in simulation. We used MoveIt! to interface with the arms and Isaac Sim for rendering. Our setup produces physically feasible plans and can directly control real robots.
>
> **Results**: We reported performance in real-world planning tasks and a scalability analysis. In the former, we reported averaged planning times in the "Runtime" column, and in the latter, we reported success rates as a proxy. We will add a plot with this data and rename "Runtime" to "Planning Time" in the tables to avoid confusion.
>
> **Challenges**: Our acceleration helps with reducing the impact of local minima on the search, though this is still challenging in clutter. Additionally, when a conflict occurs between arms when one has already been waiting at its goal for a significant amount of time, the waiting arm may be required to keep planning until after the conflict time, which is expensive in manipulation. Future work could explore approaches similar to SIPP and target-reasoning to address this challenge in the multi-arm manipulation domain.
>
> Sincerely,
>
> The Authors

---

### Official Review · Reviewer_vumT · 2024-01-23

**Significance And Importance:** 2
**Soundness:** 3
**Novelty:** 2
**Clarity:** 3
**Overall Evaluation:** 2
**Confidence:** 3

**Weaknesses:**

1: Minor weaknesses that are easily fixable.

**Contributions Of The Paper:**

This paper presents an extension of Enhanced Conflict-Based Search (ECBS) [Barer'14] for multi-agent manipulation planning that uses experience of each agent to help guide planning. The proposed method is called xECBS, while the experiences used are paths of each robot. The paper showed the complete and boundedness property of CBS is maintained. Experimental results on two different scenarios with varying number of agents, ranging from 2 to 10, indicate xECBS significantly outperform ECBS.

**Ethical Considerations:**

(1) Not Applicable: The paper does not have any ethical considerations to address

**Nomination For Best Paper:**

No

**Questions For Authors:**

Please elaborate on the effect of varying experience set to the performance of the overall algorithm. Specifically, what are the suitable criteria, perhaps in terms of coverage, for the experience set.

I've read the authors' rebuttal and am happy to keep my initial score.

**Reproducibility:**

2: Some details are missing, but the paper still appears to be replicable with some effort.

**Strengths Of The Paper:**

The use of experience in planning is interesting.
The paper has provided both theoretical and experimental justifications.
The experimental justification is relatively thorough.

**Weaknesses Of The Paper:**

I would imagine the performance of the proposed approach highly depends on the set of experience being used. But, there seems to be no mention on the effect of different sets nor on suitable criteria for this experience set.

Related to the above, given the use of experience in this paper seems quite analogous to the use of replay buffers in Reinforcement Learning, it would be useful to elaborate how the use and perhaps effects of experience in planning is similar or differ from the use of replay buffers in Reinforcement Learning.

It would be useful to provide planning time.

---

> ### Author Rebuttal · Authors · 2024-01-28
>
> Dear Reviewer vumT, thank you for your valuable feedback. We are committed to addressing your concerns to improve our paper's clarity and robustness.
>
> **Set of Experiences Used**: As you have observed, the set of experiences used helps guide the search as well as accelerate its exploration. In our experimental evaluation, we have considered three ways to form this set: (a) the previous path computed for this agent, (b) all paths computed for this agent in the CT branch, and (c) the set of all paths computed for this agent across the CT (lines 302-306). We found that (a) and (b), with narrower coverage, were more beneficial than (c), which directs the search toward potentially congested regions, often leading to creating more high-level nodes and increased workload. Comparing (a) with (b), we observed that the former provided speed-ups without the computational overhead of processing more experiences. This paper demonstrates the promise of reusing experiences in CBS-like algorithms, leaving improvements in experience selection for future work.
>
> **Reinforcement Learning**: Replay buffers store experiences encountered by an agent during interactions with an environment, typically consisting of state-action pairs, rewards, and resulting next states. These experiences are then replayed to train a policy for RL agents. As noted, our proposed experience reuse in graph search shares strong similarities with replay buffers in RL. Both leveraging previous rollouts as experiences. In RL, experiences help policy training in terms of stability and sample efficiency, and in our algorithms, experiences are directly used in the search process, potentially reducing the work needed to find a solution. We will include this similarity in the Related Work section.
>
>
> **Planning Times**: Thank you for identifying an opportunity to better communicate planning times. In our experiments, we reported performance in real-world planning tasks and performed a scalability analysis. In the former, we reported average planning times across successful trials in the "Runtime" column, and in the latter, we omitted the planning times – reporting success rates as a proxy. To clarify the results of the bin-picking and shelf-rearrangement experiments, we will rename the "Runtime" column to "Planning Time." For the scalability analysis, we will add a plot with planning time data.
>
> Sincerely,
>
> The Authors

---

### Meta-Review · Area_Chair_JEPX · 2024-02-05

**Recommendation:** Accept (Oral)
**Confidence:** 4

**Metareview:**

The paper investigates a challenging  Multi-Robot-Arm Motion Planning (MRAMP) problem and suggests a variant of the prominent multi-agent pathfinding (MAPF) algorithm, Conflict-Based Search (CBS), tailored to solve MRAMP. Specifically, the authors suggest to leverage the experience (in the form of single-arm paths) gained while solving the current MRAMP problem to speed-up the further search. The resultant variants of CBS/ECBS are proven to return optimal/bounded suboptimal solutions and are shown to notably outperform competitors while solving challenging practically-inspired tasks involving 4 and 8 armed robots (each having 7 degrees of freedom).

The main strength of the paper is that the authors show how a very challenging high-DOF planning problem can be efficiently tackled. The quality of writing is, generally, clear. The suggested methods are studied both theoretically and empirically. Overall, the paper seems to provide value to the ICAPS community (especially for those who are interested in non-trivial MAPF variants) and is worth including into the program. Addressing the concerns raised by the reviewers (including better explaining the empirical setup and the obtained results, providing more background to the readers who are not familiar with CBS-like planners, etc.) does not require a second round of reviewing and will further increase the quality of the paper.

**Ethical Considerations:**

(1) Not Applicable: The paper does not have any ethical considerations to address